# A Review on Biomedical Applications of Plant Extract-Mediated Metallic Ag, Au, and ZnO Nanoparticles and Future Prospects for Their Combination with Graphitic Carbon Nitride

**DOI:** 10.3390/ph18060820

**Published:** 2025-05-29

**Authors:** Priyanka Panchal, Protima Rauwel, Satya Pal Nehra, Priyanka Singh, Mamta Karla, Glemarie Hermosa, Erwan Rauwel

**Affiliations:** 1Institute of Veterinary Medicine and Animal Sciences, Estonian University of Life Sciences, 51006 Tartu, Estonia; glemarie.hermosa@emu.ee; 2Department of Aeronautical Engineering, Estonian Aviation Academy, 61707 Tartu, Estonia; protima.rauwel@eava.ee; 3Center of Excellence for Energy and Environmental Studies, Deenbandhu Chhotu Ram University of Science and Technology, Sonipat 131039, India; spnehra.energy@dcrustm.org; 4Faculty of Management Studies, University of Delhi, Delhi 110007, India; drpriyancka@gmail.com; 5Computer Science and Engineering Department, Deenbandhu Chhotu Ram University of Science and Technology, Sonipat 131039, India; mamtakarla21@gmail.com

**Keywords:** green synthesis, silver nanoparticles, gold nanoparticles, zinc oxide nanoparticles, graphitic carbon nitrides, nanocomposite, antimicrobial activity, antioxidant activity, cytotoxicity

## Abstract

Since the publication of the 12 principles of green chemistry in 1998 by Paul Anastas and John Warner, the green synthesis of metal and metal oxide nanoparticles has emerged as an eco-friendly and sustainable alternative to conventional chemical methods. Plant-based synthesis utilizes natural extracts as reducing and stabilizing agents, minimizing harmful chemicals and toxic by-products. Ag nanoparticles (Ag-NPs) exhibit strong antibacterial activity; Au nanoparticles (Au-NPs) are seen as a promising carrier for drug delivery and diagnostics because of their easy functionalization and biocompatibility; and ZnO nanoparticles (ZnO-NPs), on the other hand, produce reactive oxygen species (ROS) that kill microorganisms effectively. These nanoparticles also demonstrate antioxidant properties by scavenging free radicals, reducing oxidative stress, and preventing degenerative diseases. Green syntheses based on plant extracts enhance biocompatibility and therapeutic efficacy, making them suitable for antimicrobial, anticancer, and antioxidant applications. Applying a similar “green synthesis” for advanced nanostructures like graphitic carbon nitride (GCN) is an environmentally friendly alternative to the traditional ways of doing things. GCN exhibits exceptional photocatalytic activity, pollutant degradation efficiency, and electronic properties, with applications in environmental remediation, energy storage, and biomedicine. This review highlights the potential of green-synthesized hybrid nanocomposites combining nanoparticles and GCN as sustainable solutions for biomedical and environmental challenges. The review also highlights the need for the creation of a database using a machine learning process that will enable providing a clear vision of all the progress accomplished till now and identify the most promising plant extracts that should be used for targeted applications.

## 1. Introduction

In contrast to conventional chemical and physical synthesis techniques, the green synthesis of nanoparticles (NPs), particularly silver (Ag-NPs) [1,2,3,4], gold (Au-NPs) [5,6,7,8,9], and zinc oxide (ZnO-NPs) [10,11,12,13,14], has become an important and sustainable alternative [15,16,17]. The production of conventional NPs usually entails the use of hazardous chemicals and high temperatures. These processes not only take a lot of energy but also produce toxic by-products that may pollute the environment [18,19,20,21,22]. The primary advantages of green synthesis are the use of renewal resources and its capacity to generate nanoparticles with enhanced properties [23,24,25,26]. Bioactive substances in plant extracts, such as polyphenols, flavonoids, terpenoids, and alkaloids, function as reducing agents and stabilizers, facilitating the regulated and quick synthesis of nanoparticles without the need of harmful chemicals [27,28,29,30]. These phytochemical compounds improve the stability, solubility, and dispersibility of the nanoparticles, which are crucial for their properties and efficacy in many applications [31,32,33,34,35].

Moreover, the antimicrobial properties of green-synthesized nanoparticles are of interest because these nanoparticles have significant antimicrobial efficacy against a wide range of pathogens, including bacteria, fungi, and viruses [36,37,38,39,40]. Their mechanisms of action often entail the production of reactive oxygen species (ROS), resulting in oxidative damage to bacterial cell membranes, proteins, and DNA, ultimately causing cell death [38,39,40]. By minimizing dependence on synthetic chemicals, green synthesis guarantees that these nanoparticles are safer for environmental and human health. Beyond their antibacterial characteristics, Ag-NPs, Au-NPs, and ZnO-NPs also exhibit antioxidant activity. For example, Au-NPs have shown the ability to promote the production of antioxidant enzymes, including superoxide dismutase (SOD) and catalase, which mitigate detrimental free radicals, thereby safeguarding cells from oxidative injury. These qualities are exceedingly advantageous for preventive healthcare, underscoring the significance of the plant-based synthesis in enhancing human health [8,9,26,41,42,43,44,45,46,47]. Another notable benefit of green-synthesized Ag-NPs, Au-NPs, and ZnO-NPs is their potential for targeted cancer therapy, attributed to their nano size, extensive surface area, and facile functionalization with biomolecules, rendering them exemplary candidates for selective interactions with cancer cells [7,8,14,19,48,49,50,51,52,53].

Research indicates that these nanoparticles may trigger death in cancer cells by mechanisms including DNA damage, mitochondrial malfunction, and reactive oxygen species production [14,19,31]. The capacity to functionalize these nanoparticles with particular ligands such as Folic Acid, Hyaluronic Acid, Galactose, and Cholesterol-based ligands, which target cancer cells overexpressing folate receptors, facilitates targeted medication delivery, minimizing adverse effects and enhancing therapeutic success. This focused methodology is facilitated by green synthesis that can alter the surface of nanoparticles to improve their specificity and selectivity in biological applications [54,55,56]. Moreover, clinical trials have already shown that plant extract-mediated nanoparticle synthesis is a viable approach in the field of nanomedicine [57,58]. Currently, most nanomaterials used in nanomedicine are synthesized using conventional methods. However, these traditional synthesis processes are often complex and/or expensive [58,59]. In contrast, plant-based synthesis methods offer a promising, cost-effective alternative with the potential to enhance nanoparticle properties. Therefore, it is crucial to identify the most suitable plant extracts and synthesis protocols. This can be achieved through the application of machine learning tools to develop a comprehensive database. Such a database could provide insights into optimal synthesis conditions or highlight areas for improvement to ensure that plant-mediated nanoparticles match the performance of those produced by conventional methods [60,61,62].

In a similar way, graphitic carbon nitride (GCN) has become a particularly appealing material in recent years owing to its many possible applications in photocatalysis, energy storage, environmental remediation, and biomedicine. Its distinctive structure, superior thermal stability, and exceptional optical qualities make it a material of considerable worldwide interest [63,64,65,66,67]. Currently, GCN is primarily synthesized using nitrogen-rich chemical reagents such as urea, melamine, cyanamide, and dicyandiamide. In contrast, green synthesis emphasizes the use of nitrogen-rich plant-based precursors, aligning with the principles of sustainable and eco-friendly chemistry while reducing environmental impact.

## 2. Different Synthesis Methods Used for Nanoparticles Investigation

Top–Down Approach: The top–down approach to NPs synthesis involves the use of physical methods to break down bulk materials into nanoscale particles. Physical methods include ball milling, evaporation–condensation [68,69,70,71,72], arc discharge [70,73], laser ablation [70], and spray pyrolysis [71,72]. These methods offer the advantage of producing NPs with minimal chemical contamination, ensuring high-quality end products. However, these physical methods are often associated with significant drawbacks. They are typically energy-intensive and expensive due to the need for sophisticated equipment and high operational costs.

Bottom–Up approach: The chemical synthesis process for NPs synthesis involves the reduction of metal ions into zero-valent atoms and leads to the formation of NPs typically ranging from 1 to 100 nm. In this method, chemical reagents play a crucial role in reducing metal ions and promoting the growth of crystalline structures. However, it is often necessary to incorporate additional stabilizers to uphold the stability of the nanoparticles, preventing aggregation and ensuring a controlled particle size and shape, which is crucial for attaining the desired physiological properties [73,74,75,76]. The chemical synthesis process, while versatile, presents several limitations. Common chemical methods encompass conventional reduction, sol-gel processes [77], chemical vapor deposition [78], reverse micelle synthesis, co-precipitation, electrochemical reduction, and solvothermal methods [79]. Conversely, green synthesis utilizes living organisms and bio-derived compounds, including plant extracts, microorganisms (such as bacteria, fungi, and viruses), and enzymes to generate nanoparticles [80].

This approach presents notable benefits compared to conventional chemical methods, as it eliminates the need for hazardous substances, promotes more eco-friendly processes, and can result in nanoparticles that exhibit enhanced biocompatibility and stability [81,82,83]. The approach provides streamlined, secure, and eco-conscious methods while preserving oversight of nanoparticle properties, leading to its growing preference in biomedical applications and various other sectors [84,85,86,87,88,89,90,91], as shown in Figure 1 and Table 1.

### Merits of Biological Methods for Sustainable Approach

Biological methods utilize natural resources, like plant extracts [88], bacteria [72,88], fungi, or algae [22,72,88], that act as reducing and stabilizing agents. They eliminate the use of hazardous chemicals, reducing environmental pollution [92,93,94,95,96]. Their non-toxicity makes them highly biocompatible, minimizing side effects and making them ideal for therapeutic applications like antibacterial and anticancer treatments [95,96,97,98]. In terms of cost-effectiveness, since biological approaches need softer reaction conditions, including ambient temperature and standard pressure, it therefore reduces energy consumption, thus making it cost-effective and more environmental friendly (i.e., carbon footprint). In fact, plant-based synthesis is particularly cost-effective, as easily accessible raw materials such as leaves, stems, and fruits [99,100,101,102]. Biological synthesis is also highly scalable and simple, requiring minimal equipment compared to physical and chemical methods. Moreover, the nanoparticles produced through this approach exhibit enhanced stability, as natural capping agents like proteins, carbohydrates, and polyphenols prevent aggregation and ensure prolonged efficacy [102,103,104,105,106,107,108,109,110]. Lastly, these methods support sustainability by reducing reliance on toxic chemical precursors, instead utilizing plant materials and agricultural waste, thereby lowering environmental risks and promoting a circular economy [111,112,113,114,115,116,117,118,119,120,121,122].

## 3. Methodology of Plant-Based Synthesis of Nanoparticles and Their Mechanisms

Many plant parts (such as fruits, leaves, stem, roots, and flowers) are rich in bioactive compounds like polyphenols, flavonoids, and alkaloids, which can act as both reducing and stabilizing agents [115,116,117,118,119,120,121,122,123,124].

Preparation of plant extract: The plant material is washed, dried, and chopped. The chopped material is boiled in distilled water to extract the bioactive compounds [125,126,127,128]. Addition of Metal Salts: A solution of metal salts (e.g., silver nitrate for Ag-NPs, gold chloride for Au-NPs, and zinc nitrate for ZnO-NPs) is prepared, preferably in distilled water. The biological extract is incorporated into the metal salt solution. The bioactive chemicals in the extract reduce metal ions to metal atoms, hence starting the formation of nanoparticles, starting by nucleation [129,130,131,132].

Reduction Process: The bioactive molecules in the plant extract act as reducing agents. They reduce metal ions (e.g., Ag⁺, Au^3^⁺, Zn^2^⁺) to form metal nanoparticles (e.g., Ag^0^, Au^0^, Zn). The reduction process typically results in a color change in the solution, indicating nanoparticle formation [133,134,135,136,137].

Stabilization and Size Control: The bioactive compounds in the extract do not only reduce the metal ions but also stabilize the NPs by preventing agglomeration. Stabilizing agents such as proteins, polysaccharides, and other compounds are responsible for controlling the size and preventing the particles from clustering together [138,139,140,141].

Purification: Following nanoparticle synthesis, the solution is often processed using centrifugation, filtering, or dialysis to eliminate surplus biological material, unreacted metal ions, and other contaminants. The pure nanoparticles may then be dried and characterized for further investigations, as seen in (Figure 2a,b) [142,143,144,145,146,147].

### 3.1. Characterization Methods to Control the Stability and the Properties

The NPs are characterized to determine their size, shape, surface charge, and chemical composition. Some common characterization methods include UV-Vis Spectroscopy to observe the characteristic surface plasmon resonance (SPR) peak, indicating NPs formation; transmission Electron Microscopy (TEM) to examine the size, shape, and morphology of the NPs; Scanning Electron Microscopy (SEM) to analyze surface structure; X-ray Diffraction (XRD) to determine the crystalline structure and phase, and to estimate the size using Scherrer equation; Fourier Transform Infrared Spectroscopy (FTIR) to identify the functional groups responsible for the reduction and stabilization.

UV-Vis spectrophotometry shows that particle size influences absorption, with smaller nanoparticles causing a blue shift (shorter wavelengths) and larger ones causing a red shift (longer wavelengths). For example, Au-NPs of around 10 nm exhibit absorption peaks near 520 nm, whereas particles larger than 80 nm shift the peak towards 600 nm due to increased scattering and plasmon resonance effects. Shape also influences the absorption, as spherical nanoparticles exhibit a single absorption peak, while anisotropic shapes like nanorods show multiple peaks at different wavelengths [5,7,21,49,65,106,147]. Aggregation of nanoparticles leads to plasmonic coupling, resulting in a red shift. Additionally, the refractive index of the surrounding medium affects the peak position, with a higher refractive index causing a red shift and a lower one a blue shift. Ag-NPs, Au-NPs, and ZnO-NPs exhibit distinct absorption peaks between 400 and 430 nm, 520 and 550 nm, and 375 and 400 nm, as seen in Figure 3a–c and Table 2. These UV-vis results serve as robust evidence for confirming the formation of green-synthesized Ag, Au, and ZnO NPs.

FTIR spectroscopy analysis of Ag-NPs, Au-NPs, and ZnO-NPs synthesized from plant extracts reveals the presence of key functional groups. In the case of Ag-NPs synthesized from banana pulp [55], *Cassia tora* seed [95], *Jasminum nudiflorum* flower [109], Sesbania grandiflora leaves [124], *Vaccinium arctostaphylos* fruit [129], and peel, extracts show peaks at 820 C-H vibrations of cellulose, 502 C-H bending vibration, Ag-O bonds,1635 to 1645 cm⁻^1^ ketone group due to -C=O stretching alkynes due to C-C stretching, and alcohol or phenol groups due to -OH stretching in the bands at 2134 cm⁻^1^ and 3317 cm⁻^1^, respectively, as shown in (Figure 4a). Similarly, Au-NPs, commonly reveal peaks appear at 3200–3400 cm⁻^1^ (O-H stretch) hydroxyl groups, 1600 cm⁻^1^ (C=O stretch) carbonyl groups, 1400–1500 cm⁻^1^ (C-H bending) aromatic or aliphatic groups, 1000–1200 cm⁻^1^ (C-O stretch) alcohols, phenols, ethers, and 600–800 cm⁻^1^ (C-H bending) aromatic rings, corresponding to different plant extracts such as *Curcuma pseudomontana* flower [9], *Glaucium flavum* leaves [15], *Citrus* peel [32], and *Delphinium chitralense* tuber [82], as shown in (Figure 4b). Moreover, ZnO-NPs synthesized from *Lantana camara* flower [24], *Lepidium sativum* seed [31], *Ipomoea sagittifolia* Burm.f leaves [53], *Olive* fruit [86], and *Punica granatum* peel [130] typically appear at 3400–3500 cm⁻^1^ (O-H stretch), 1600 cm⁻^1^ (O-H bending), 500–700 cm⁻^1^ (Zn-O stretch), and 1100–1300 cm⁻^1^ (C-O stretch), indicating the presence of hydroxyl groups, water, and capping agents involved in stabilization, as shown in (Figure 4c). The findings underscore the importance of FTIR to identify the essential function of plant-derived bioactive compounds, including flavonoids, phenols, and proteins on the surface of green-synthesized nanoparticles, hence enabling their many biological and environmental uses [17,37,71].

XRD study is a key routine method to characterize the crystalline structure and phase purity of Ag-NPs, Au-NPs, and ZnO-NPs produced by plant extracts and the estimation of the crystallite size, as seen in (Figure 5a–c). Distinct diffraction peaks for Ag-NPs were observed at 37–38° (111), 44–47° (200), 63–64.5° (220), and 77–79° (311), confirming the face-centered cubic structure, as evidenced in various studies involving extracts from Abelmoschus esculentus flower [27], *Withania coagulans* seed [35], banana pulp [54], *Carica papaya* leaf [92], and *Rosa canina* fruit [122]. Similarly, Au-NPs display characteristic peaks at 2θ values corresponding to 37–39° (111), 44–45° (200), 64–67° (220), and 77–78° (311), confirming the face-centered cubic crystalline structure in different plant extract-mediated Au-NPs, such as *Capsicum chinense* Leaves, stem and root [2]; *Flaxseed* seed [7]; *Jatropha integerrima* Jacq. flower [20]; and *Citrus* peel [32]. In addition, for XRD, peaks for ZnO-NPs typically appear at 28–32° (100), 34–35° (002), 36–39° (101), 44–48° (102), 56–57° (110), 62–63° (103), 66–67° (112), 72–73° (004), and 77–78° (220), confirming the hexagonal wurtzite structure with various reported studies, such as *Phlomis* leaf [29], *Andrographis alata* whole plant [13], Plantain peel [34], *Caesalpinia crista* seed [44], and *Capparis spinosa* fruit [52].

The XRD findings conclusively demonstrate the successful production of crystalline nanoparticles with unique and stable stuctures, hence improving their efficacy in biological, catalytic, and environmental applications [8,36,93,135,137].

### 3.2. Morphology and Size Distribution

SEM + EDS (morphology and elemental distribution), TEM and DLS (morphology and size), and zetapotential (stability) can provide valuable insights information of plant extract-synthesized Ag-NPs, Au-NPs, and ZnO-NPs, as shown in Figure 6I(a–f). In the case of Ag-NPs, they show predominantly spherical, regular, circular shapes, and sizes with high stability, such as those synthesized from *Eucalyptus globulus* and *Salvia officinalis* (Spherical) [4], *Aervalanata* (Spherical) [17], *Areca catechu* nut (Regular spherical) [21], *Allium cepa* (spherical) [30], *Ferula asafoetida* (Circular) [101], and *Ocimumcanum* (Rod and Spherical) [115], demonstrating well-dispersed and stable nanoparticles. For Au-NPs, they mainly show spherical, hexagonal, and circular particles like in the case of studies on *Crataegus oxyacantha* (Spherical) [1], flaxseed (Spherical and Triangular) [7], *Jatropha integerrima* Jacq. (Spherical) [20], *Licorice* (Circular) [25], and saffron (Spherical and Oval) [51]. Similarly, ZnO-NPs synthesized from *Andrographis alata* (Spherical, Oval, and Hexagonal) [13], *Berberis aristate* (Needle) [23], *Cymbopogon citratus* (Spherical) [54], *Phlomis* leaf (Hexagonal) [85], *Punica granatum* peel (Spherical and Hexagonal) [90], and *Capsicum chinense* (Spherical) [94] extracts revealed a variety of shapes, including spherical, irregular, and hexagonal, with a strong influence of plant compounds in controlling nanoparticle morphology during the growth. These SEM and TEM observations underscore the effectiveness of plant-mediated green synthesis methods in controlling the size, shape, and uniformity of nanoparticles, which are crucial for their diverse applications in antimicrobial, antioxidant, catalytic, and biomedical fields. According to reported studies, particle size can vary depending on the characterization technique used, such as XRD, DLS, or TEM.

EDS is used to analyze the elemental composition of nanoparticles. Ag-NPs strong peak at approximately at 3.0 keV (Ag L_α_); Au-NPs 2.2 keV (Au M_α_), with additional peaks at 9.7 keV (Au L_α_) and 11.4 keV (Au L_β_); and ZnO-NPs peak at 1.0 keV (Zn K_α_) are visible, as shown in Figure 6II(a–c) [4,7,54]. Zeta potential is used to evaluate the stability and dispersion of nanoparticles. However, Ag-NPs and Au-NPs exhibit values between −20 and −40 mV under pH 7–9, and ZnO-NPs range from −15 to −30 mV under pH 9–10. These values indicate stable, well-dispersed nanoparticles and good colloidal stability, as shown in Figure 6II(d–f) [4,7,20,21,29,54]. Moreover, DLS was used to determine the hydrodynamic size distribution of Ag-NPs, Au-NPs, and ZnO-NPs synthesized using a plant extract. This technique provides valuable insight into the dispersion quality and stability of the nanoparticles in solution, essential for biological and catalytic applications, as shown in Figure 6II(g–i) [6,100,130]. However, the main limitation using DLS is in the case of agitated nanoparticles or very small nanoparticles (<50 nm), which is usually the case for nanoparticles synthesized by plant extract methods.

## 4. Literature Study

During the last 10 years, many studies reported the synthesis of Ag-NPs, Au-NPs, and ZnO-NPs using plant extract-based synthesis. Ag-NPs and ZnO-NPs are known for their strong antibacterial properties, mainly attributed to their capacity to produce reactive oxygen species (ROS) under light exposure that interfere with microbial cell membranes and DNA. Au-NPs nanoparticles are more known for their compatibility with biological systems and their use in drug delivery and diagnostic applications. These nanoparticles demonstrate significant antioxidant activity, effectively scavenging free radicals and lowering oxidative stress, which positions them as beneficial agents in the fight against degenerative diseases. Studies consistently emphasize the environmentally friendly, cost-effective, and versatile nature of green-synthesized NPs, showcasing their immense potential for applications in biomedicine, environmental remediation, and other fields, as illustrated in Table 3, Table 4 and Table 5.

### 4.1. Pioneering Studies on Ag-NPs Derived from Plant Extracts

As shown in Table 3, various plant part extracts have been used to synthesize Ag-NPs, which have emerged as a pivotal innovation in nanotechnology, particularly due to their invaluable biomedical applications. *Euphorbia serpens* plant (leaves) extract-mediated Ag-NPs demonstrated strong antibacterial activity, particularly against *E. coli* (20 ± 06 mm) and *S. typhi* (18 ± 0.5 mm), with moderate antifungal effects against *C. albicans* and *A. alternata*. They also showed significant cytotoxicity against Artemia salina with LD_50_ values of 5.37 and 5.82 at 1–2 μg/mL concentrations [2]. Additionally, Aervalanata (flower) extract biosynthesized Ag-NPs exhibited antibacterial activity against both Gram-positive and Gram-negative bacteria, and demonstrated antioxidant potential at a concentration of 100 µg/mL in the DPPH radical scavenging assay [17]. Similarly, Ag-NPs Salvia officinalis plant extract exhibited antimicrobial properties, both Gram-positive and Gram-negative bacteria (four strains), and the antimicrobial properties were assessed using multiple methodologies [143]. Notably, *Allium cepa* L. (bulb) extract-mediated Ag-NPs exhibited strong antibacterial activity against *P. aeruginosa*, *B. subtilis*, and *S. aureus* strains, outperforming standard antibiotics. Their antioxidant properties were demonstrated through various assays, with IC_50_ values of 116.9 µg/mL in the β-carotene linoleic acid assay, and also showed inhibitory effects on acetylcholinesterase and butyrylcholinesterase enzymes [30]. Ag-NPs produced using banana (pulp) extra exhibited strong antibacterial activity, especially against *E. coli*, and enhanced voltage regulation in bio-electrochemical cells (BEC) t. Ag-NPs investigated through Flaxseed extract demonstrated significant antimicrobial activity against *A. baumannii*, with a minimum inhibitory concentration of 2 μg/mL, showing no toxicity at lower concentrations and enhanced effects compared to chemically synthesized Ag-NPs, though cytotoxicity increased at higher concentrations [100]. The antioxidant assessment revealed superior antioxidative activity (84.48%) in green-synthesized *Morinda lucida* (leaves) nanoparticles compared to chemically synthesized ones (75.87%). Additionally, the nanoparticles exhibited dose-dependent anti-inflammatory effects, with IC_50_ values of 30.19 μg/mL for plant extract-mediated Ag-NPs and 34.14 μg/mL for chemically synthesized Ag-NPs. The lower IC50 value of plant extract-mediated NPs (30.19 μg/mL) indicates stronger and more effective activity, likely due to the presence of phytochemicals in the green synthesis [110]. *Otostegia persica* phyto-fabricated Ag-NPs exhibited superior antioxidant activity and significant antibacterial effects against both Gram-positive and Gram-negative bacteria. They also demonstrated dose-dependent anti-inflammatory and antifungal properties, with MIC values of 18.75, 37.5, and 75 µg/mL against *Candida* species, making them a promising agent for medical treatments [117]. Furthermore, *Terminalia chebula* (fruit) extract-synthesized Ag-NPs exhibited strong antibacterial and antioxidant activities, with no significant toxic effects on zebrafish embryos, highlighting their potential as therapeutic agents [127]. However, Table 2 summarizes their shape, size, and potential for different biomedical applications.

### 4.2. Studies on Au-NPs Synthesized from Plant Extract

These studies highlight the potential and application of Au-NPs synthesized via green methods, demonstrating their broad biological, catalytic, and medicinal applications. *Crataegus oxyacantha* (twig) extract-mediated Au-NPs exhibited strong urease inhibitory activity (99.25% inhibition, IC_50_ = 1.38 ± 0.3), surpassing the standard thiourea, highlighting their potential for biomedical applications in homeopathic and pharmaceutical industries [1]. Au-NPs synthesized from *M. oleifera* (leaves) extract exhibited low cytotoxicity and promoted neuronal cell regeneration in animal model studies [6]. *Curcuma pseudomontana* (flower) extract-mediated Au-NPs exhibited strong antibacterial activity against *P. aeruginosa*, *S. aureus*, *B. subtilis*, and *E. coli*, with a maximum inhibition efficiency of 94% in the human red blood cell method. They also demonstrated significant antioxidant and radical scavenging activities, highlighting their potential for biomedical applications [9]. Au-NPs exhibited strong antimicrobial activity against *E. coli*, *P. aeruginosa*, *S. aureus*, and *B. subtilis*, along with effective antioxidant activity (IC_50_ = 94.5 mg/mL) synthesized from *Kaempferia parviflo* (rhizomes) [26]. *Cynodondactylon* (grass) extract-loaded Au-NPs exhibited cytotoxicity against the MCF-7 cell line, with an IC_50_ of 31.34 μg/mL, and induced ROS generation, DNA fragmentation, and mitochondrial membrane changes. They also demonstrated significant antibacterial activity against clinically isolated pathogens like *Enterobacter cloacae*, *Staphylococcus haemolyticus*, and *Staphylococcus petrasiisubsp* [39]. Au-NPs exhibited potent dose-dependent anticholinesterase activity, with IC_50_ values of 15 ± 0.42 μg/mL for AChE and 12 ± 0.42 μg/mL for BChE synthesized from *Delphinium chitralense* (tuber) [82]. However, Table 4 summarizes their shape, size, and potential biomedical applications.

### 4.3. Exploratory Studies on ZnO-NPs Produced from Plant Extracts

ZnO-NPs synthesized using Ailanthus altissima (leaves) extract-mediated ZnO-NPs showed dose-dependent antibacterial and antioxidant activity at 20 mg/mL ZnO-NPs; the maximum bactericidal potential of ZnO-NPs was reported against *Staphylococcus aureus* (201.2 mm). These ZnO-NPs have also an IC_50_ value of 78.23 µg/mL, indicating that they are an effective antioxidant [11]. *Mucuna pruriens* (peel) extract-mediated ZnO-NPs exhibited selective cytotoxicity against HeLa and HEK 293 cancer cell lines in the MTT assay, with no toxicity to normal cells. This study presents a simple, non-toxic biosynthesis method for ZnO-NPs with promising anticancer and antioxidant applications [14]. ZnO-NPs synthesized from *C. spinosa* L. (fruit) extract showed strong antioxidant activity, low cytotoxicity on L929 fibroblast cells, and good RBC biocompatibility. With no hemolytic effects at 7.5–120 µg/mL, they offer a safe alternative for pharmaceutical and biomedical applications [52]. *Myrica* esculenta (Fruit) extract-fabricated ZnO-NPs exhibited superior antimicrobial activity (MIC: *B. subtilis* 0.031 mg/mL, *S. aureus* 0.062 mg/mL, *P. aeruginosa* 0.125 mg/mL). They also showed strong concentration-dependent antioxidant activity (IC_50_-DPPH: 182.63 µg/mL; FRAP: 129.44 µM FeSO_4_ equivalents) [93]. Biosynthesized ZnO-NPs exhibited cytotoxic activity against HEK 293 and HeLa cell lines and showed superior antioxidant potential in the DPPH assay compared to conventional ZnO-NPs. These findings highlight *C. chinense* (fruit) extract as an effective agent for ZnO-NP biosynthesis, with anticancer and antioxidant properties [94]. Boerhaavia *erecta* (leaves) extract phyto-fabricated ZnO-NPs inhibited advanced glycation end products (AGEs) formation by blocking Amadori products, trapping dicarbonyl intermediates, and breaking glycated protein cross-links. Additionally, they protected RBCs from MGO-induced damage, highlighting their potential in diabetes-related complications [104].

### 4.4. Biomedical Applications

#### 4.4.1. Antimicrobial and Antioxidant Activities

The increasing resistance of bacteria such as *Bacillus subtilis*, *Staphylococcus aureus*, *Vibrio cholerae*, *Escherichia coli*, *Pseudomonas aeruginosa*, *Aeromonas hydrophila*, and *Proteus mirabilis* [133] to several commonly used antibiotics has become a critical worldwide medical concern that requires immediate solutions. Ag-NPs, Au-NPs, and ZnO-NPs have emerged as promising alternatives in the field of biomedical applications. Research showed that plant extracts-mediated NPs can effectively destroy harmful bacteria and inhibit their growth. Table 3, Table 4 and Table 5 summarize published studies that report the green synthesis of nanoparticles using plant extract and which parts of the plants that were used. In these studies, almost all the plant extracts-mediated NPs exhibited significant antimicrobial effects against the tested microorganisms. Mechanism: As shown in Figure 7a, antibacterial mechanisms of plant extracts-mediated NPs operate via multiple pathways. These include the generation of ROS, which disrupts bacterial membranes, damages proteins, and induces oxidative stress in bacterial DNA. This mechanism is known to be the main mechanism involved in ZnO-NPs. In the case of Ag-NPs, this is the release of Ag⁺ ions that bind to thiol groups in bacterial enzymes, inhibiting essential metabolic functions. The release of Zn^2^⁺ ions by ZnO-NPs is lower, but Zn^2^⁺ ions disrupt bacterial ion homeostasis and catalyze additional ROS production under light exposure. The NPs attach to bacterial membranes via electrostatic interactions, leading to membrane disruption, leakage of cytoplasmic contents, and ultimately resulting in cell death. These NPs inhibit protein synthesis through binding to ribosomal subunits and impede biofilm formation by disrupting bacterial adhesion and the extracellular matrix. Furthermore, phytochemicals that originate from plant extracts work in conjunction with nanoparticles, improving their stability, bioavailability, and overall effectiveness against microbes [51,130,135]. However, some studies showed that the plant extract method can also shield the production of ROS if green-synthesized nanoparticles are synthesized using beta-diketonate precursors like zinc acetate [134], and nitrate should be preferred for the synthesis using plant extract.

#### 4.4.2. Anticancer Activities

Ag-NPs, Au-NPs, and ZnO-NPs have shown potential application in cancer therapy owing to their unique properties and mechanisms of action.

Mechanism: As shown in Figure 7a, Ag-NPs, Au-NPs, and ZnO-NPs demonstrate notable anticancer mechanisms due to their high surface-area-to-volume ratios, nanoscale properties, and functionalization with plant-derived bioactive compounds. Anticancer mechanism of green-synthesized Ag-NPs, Au-NPs, and ZnO-NPs are diverse and have been extensively studied. Three primary mechanisms are often highlighted: (1) interference with cell membranes, (2) ROS-induced apoptosis, and (3) disruption of the chemistry of proteins and DNA. Plant extracts-mediated NPs interact with cancer cells in several ways, and their surface properties play a critical role in cellular internalization. The positive charges on green-synthesized NPs interact with the negatively charged phosphate groups on the lipid bilayer of cell membranes, facilitating their uptake into cells through endocytosis. Once inside, NPs accumulate in the mitochondria, where they cause structural damage, impair the electron transport chain, and activate NADPH oxidases, leading to mitochondrial dysfunction and ROS generation. ROS production is a major factor behind the anticancer effects of NPs. These elevated ROS levels induce oxidative stress, resulting in DNA fragmentation, mitochondrial dysfunction, and eventual cell death through apoptosis, autophagy, or necroptosis. The oxidative stress caused by NPs depletes cellular antioxidants like glutathione and disrupts the mitochondrial membrane potential, triggering the release of apoptotic molecules such as cytochrome-c, apoptosis-inducing factor, and endonuclease G into the cytosol, leading to cell death.

In addition to their ROS-induced apoptosis, NPs can interfere with the metabolic activity of cancer cells. They have been shown to inhibit DNA biosynthesis by directly binding to DNA and reducing the expression of DNA repair proteins, thus enhancing the anticancer effect. Beyond the ROS generation and mitochondrial dysfunction, NPs also lead to the activation of endoplasmic reticulum stress responses. The complex relationship of ROS-mediated damage, ion release, and biofilm disruption positions green-synthesized Ag, Au, and ZnO NPs as highly effective agents against oxidative stress and bacterial pathogens, highlighting their considerable promise in both biomedical and environmental fields [7,14,31,92,131].

## 5. Future Outlook of Hybrid Nanocomposites with Graphitic Carbon Nitride (GCN)

Graphitic Carbon Nitride is a metal-free semiconductor material composed of C and N compounds, known for its unique layered structure and promising candidacy for different applications in various fields such as biomedical, photo-catalysis, energy conversion, and environmental remediation. Its structure resembles graphite, with nitrogen atoms integrated into the carbon frame work, offering excellent stability. Moreover, to date, Graphitic carbon nitride traditionally has been synthesized by thermal polymerization of nitrogen-rich precursors like urea, thiourea, and melamine for huge environmental applications. However, GCN and their nanocomposites have gained attention for dye degradation, demonstrating enhanced photocatalytic activity for methylene blue. These studies used conventional chemical methods to synthesize both GCN and ZnO-NPs [63,64,144]. GCN-ZnO-NPs hybrid nanocomposites have also been studied for possible applications in drug delivery, making it a promising candidate for breast cancer treatment [146,148]. GCN has already been combined with Ag-NPs using the ex-situ method through the use of ultrasonic bath [149] and through the in-situ method [150]. The optical properties of the nanocomposite were enhanced due to the plasmonic effect of Ag-NPs. These hybrid nanocomposites were tested for the catalytic reduction of 4-nitrophenol and the degradation of 2,4-Dichlorophenoxyacetic acid. In both cases, the combination of GCN with Ag-NPs showed enhanced photocatalytic properties. GCN-Ag-NPs hybrid nanocomposites were also studied for the degradation of dyes and H_2_ production. The study showed that GCN-Ag-NPs hybrid nanocomposite exhibits superior photocatalytic activity [151]. However, in all these studies, Ag-NPs and GCN were prepared using conventional chemical methods [143,151]. Au NPs were combined with GCN and carbon nanotubes (CNTs) to study their photocatalytic properties applied to the effective photodegradation of organic pollutants and possible application in photoelectrochemical water splitting [152]. The authors showed that GCN-Au-NPs hybrid nanocomposite exhibits photocatalytic activity increased by a factor of almost 40 in the case of Rhodamine B photodegradation under visible light exposure. GCN-Au-NPs hybrid nanocomposite has been also studied for the development of fluorescent sensor for Fe^3+^ and dichromate ions detection in aqueous medium [153].

All these studies highlight the high potential to combine metal (oxide) nanoparticles with GCN. However, to date, graphitic carbon nitride has not been synthesized using plant extracts. Current investigations indicate that this approach holds significant potential for the future, similarly to metallic nanoparticles. With ongoing advancements, the effective formulation of plant extract-mediated GCN-NPs may pave the way for innovative, sustainable approaches in biomedical applications (see Table 6), pollution management, energy generation, and environmental restoration [140,141,142,143,144].

### Our Current Research Approach and Innovations

Impressed by the significance of plant-based nanoparticles and their nanocomposites, we synthesized plant-based GCN using *Ocimum tenuiflorum*, as shown in Figure 8. The plant-based GCN was derived by using *Ocimum tenuiflorum* leaves that had been kept in a hot-air oven at a temperature of 70 °C for 36 h. The dry leaves were then ground into a powder. The powder was then added to a closed ceramic crucible and heated in a muffle furnace at 550 °C for 3 h at a heating rate of 2 °C/min. GCN-NPs pellets were crushed into a fine powder with a mortar and pestle. The synthesized GCN-NPs was confirmed by the XRD. GCN-NPs exhibit a strong XRD peak at 2θ of 27.7° that corresponds to the (002) plane that is common to all GCN phases, which therefore confirms the Graphitic stacking of C_3_N_4_ in GCN-NPs. The next step is to combine plant-based synthesized GCN with metal (oxide) nanoparticles and study these hybrid nanocomposites fully produced using environmental friendly method of syntheses [155].

## 6. Limitations of Green-Synthesized Nanoparticles

However, green synthesis is not without limitations. Its scalability and reproducibility can be affected by factors such as the regional and seasonal variability of plant materials, as well as differences in extract composition based on geographic origin. Also, we face the problem of the control of synthesis with a sharp size distribution, reproducibility due to possible variation of the plant extract composition depending on the origin. For this reason, the chemical method must still be considered for synthesis. These factors hinder, increase costs, and challenge the overall feasibility of industrial production [154,156,157,158]. Also, there is the presence of chlorine in plant extracts that can induce the formation of metal chlorine like AgCl as secondary phase during the synthesis process [98]. The nature of the plant not only influences the properties of the nanoparticles through functionalization by the bioactive compounds present in the extract but also affects their morphology and size during the synthesis process [156]. While many studies have shown that locally accessible plants like cotton leaves and Coffea arabica can serve as effective agents for nanoparticle synthesis, practical limitations exist. Many plants in green synthesis produce supplementary by-products that require additional treatment before they can be used, adding extra steps, time, and expense to the process. Additionally, a major scientific hurdle is the lack of a clear understanding of the chemical pathways involved in the synthesis process. While some studies have demonstrated the effects of plant extracts on nanoparticle formation, the exact mechanisms remain largely unknown. This limits the ability to control and optimize the synthesis, making it difficult to predict outcomes or scale the process reliably. Current research indicates large variations in particle diameter, which presents a major obstacle for industrial applications. Moreover, the characterization techniques used to evaluate the synthesized nanoparticles are sometimes insufficient to understand. Without standardized protocols and deeper insight into the reaction mechanisms, achieving consistent, high-quality products suitable for commercial use remains a challenge. Therefore, while green synthesis has tremendous potential, addressing these issues related to raw material sourcing, reaction conditions, process understanding, and product quality control is essential for its transition from laboratory research to viable industrial technology [57,58,59,60,61,62,154,156,157,158].

## 7. Conclusions

In this review, we systematically explore the synthesis approaches, significance, design, advancements, and biomedical applications of plant extract-mediated nanoparticles, with a specific focus on Ag-, Au-, and ZnO-NPs. The first section highlights the importance of nanomaterials in biomedical applications, and their various synthesis methods, followed by a discussion on the significance of plant extract-mediated nanomaterials and their properties. Therefore, different plant extracts were used in the study for the nanoparticle synthesis; however, no real studies were performed to identify a specific plant family that would be best to target a specific pathogen. So, it appears necessary to use machine learning software to identify such a pattern through the creation of a database. Such studies will shed a light on the best plant extract that must be used for a specific targeted application. The second section discusses the antibacterial, antioxidant, and anticancer properties of Ag-NPs, Au-NPs, and ZnO-NPs and highlights their potential biomedical applications. In the third section, we provide key insights and future perspectives on combining these nanoparticles with GCN to enhance their properties. This shows that the similar way of synthesis involving plant extracts should be applied to the synthesis of GCN to promote better sustainability in the production of these hybrid nanocomposites. Finally, we present our recent findings and research progress on plant-based synthesized GCN-NPs, highlighting their potential for future applications in biomedical applications and environmental remediation.

This review tries to highlight the recent progresses in the field and how it is important to now combine different nanomaterials to obtain a synergistic effect to obtain new or improved properties. Additionally, there is a clear need of studies using machine learning software to identify a pattern or the most promising plant extract that must be used for a targeted application. This identification can only be accomplished through the creation of a database using such software combined with artificial intelligence.

## Figures and Tables

**Figure 1 pharmaceuticals-18-00820-f001:**
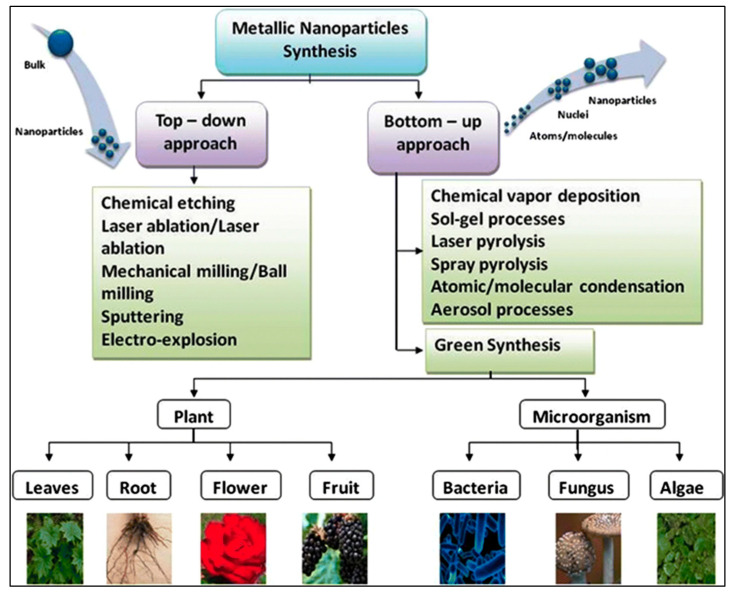
Different methods used for metallic nanoparticles synthesis [22].

**Figure 2 pharmaceuticals-18-00820-f002:**
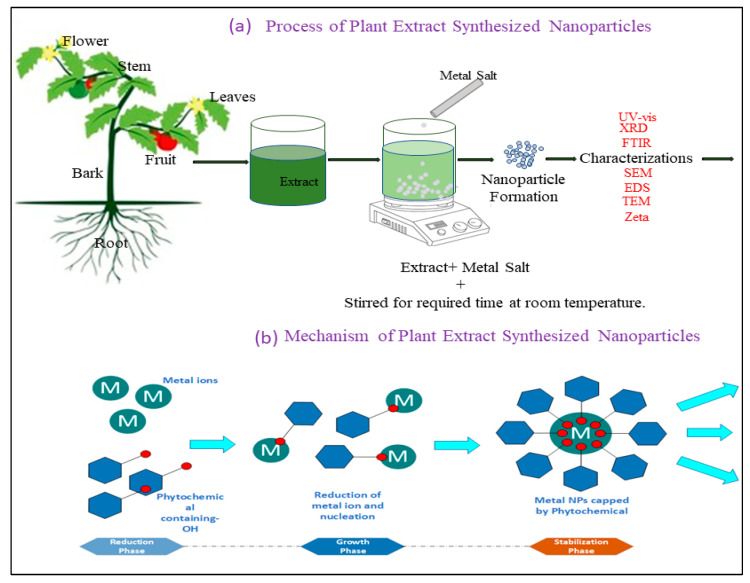
Process and mechanism of plant extract-synthesized nanoparticles.

**Figure 3 pharmaceuticals-18-00820-f003:**
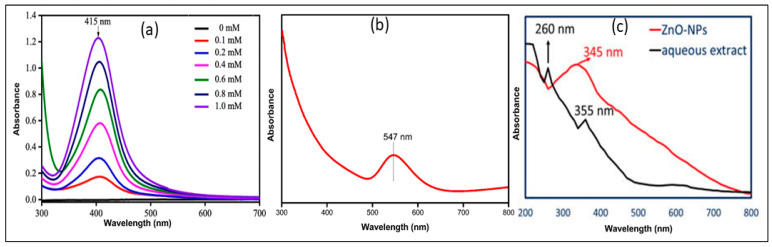
UV-vis analysis of plant extract-synthesized Ag-NPs (**a**) [21]; Au-NPs (**b**) [20]; and ZnO-NPs (**c**) [130].

**Figure 4 pharmaceuticals-18-00820-f004:**
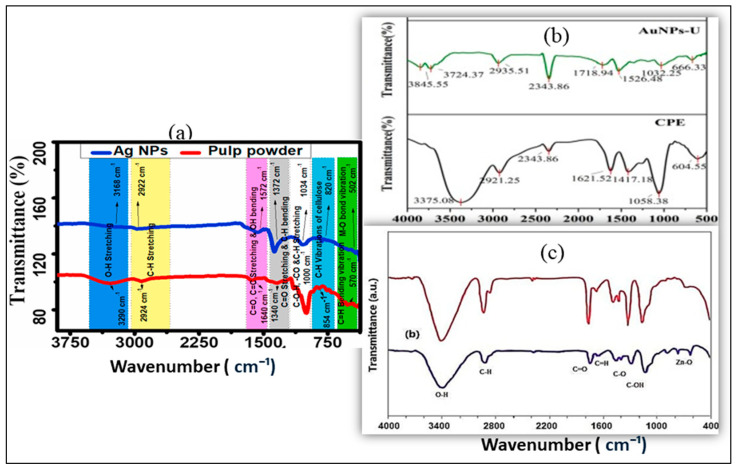
FTIR analysis of plant extract-synthesized Ag-NPs (**a**) [55]; Au-NPs (**b**) [32]; and ZnO-NPs (**c**) [130].

**Figure 5 pharmaceuticals-18-00820-f005:**
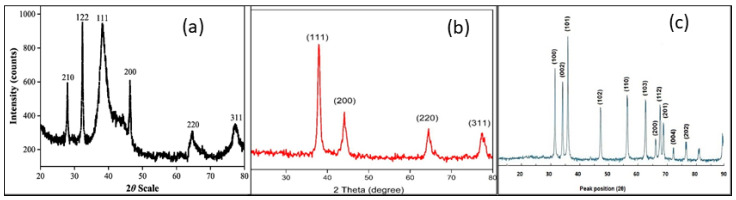
XRD analysis of plant extract-synthesized Ag-NPs (**a**) [21]; Au-NPs (**b**) [20]; and ZnO-NPs (**c**) [29].

**Figure 6 pharmaceuticals-18-00820-f006:**
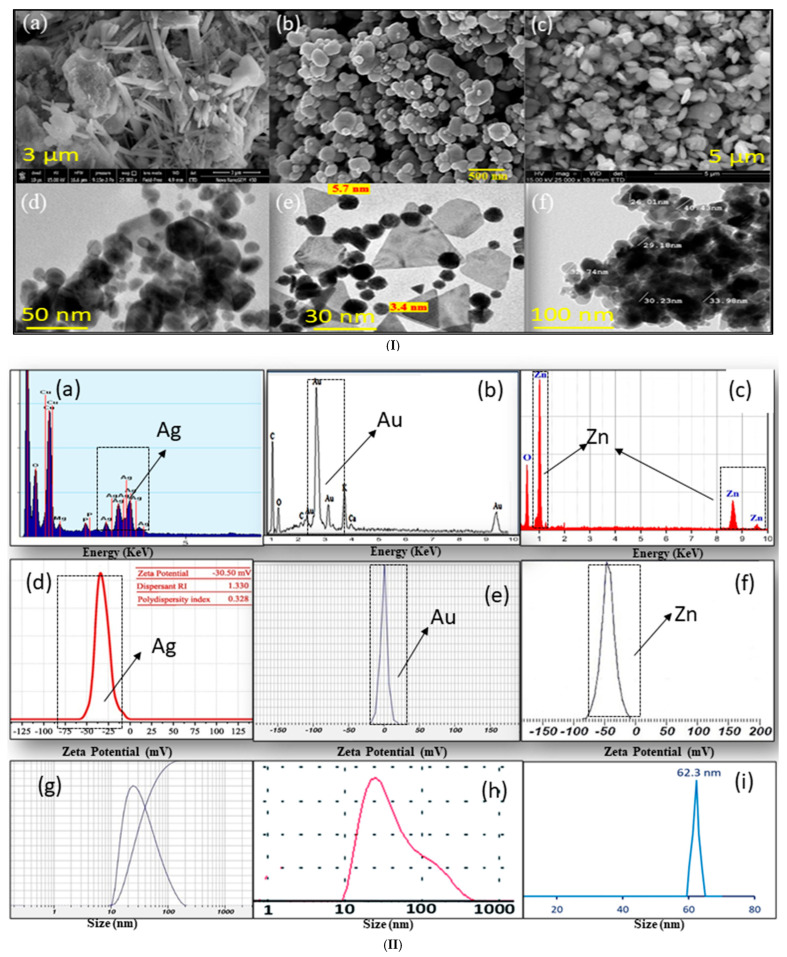
(**I**) SEM and TEM studies of plant extract-synthesized Ag-NPs ((**a**) [115], (**d**) [4]); Au-NPs ((**b**) [25], (**e**) [7]); and ZnO-NPs ((**c**) [94], (**f**) [90]). (**II**) EDS, zeta Potential, and DLS analysis of plant extract-synthesized Ag-NPs ((**a**) [4], (**d**) [21], (**g**) [100]); Au-NPs ((**b**) [7], (**e**) [20], (**h**) [6]); and ZnO-NPs ((**c**) [54], (**f**) [29], (**i**) [130]).

**Figure 7 pharmaceuticals-18-00820-f007:**
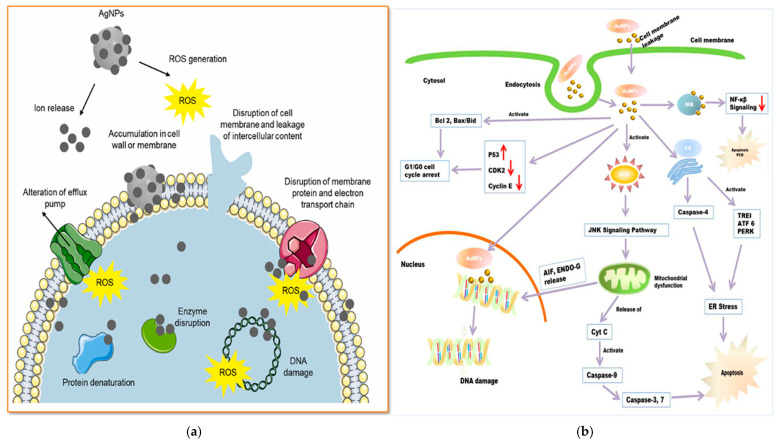
Reported antibacterial (**a**) and anticancer (**b**) mechanisms of plant extract-mediated NPs [19,135].

**Figure 8 pharmaceuticals-18-00820-f008:**
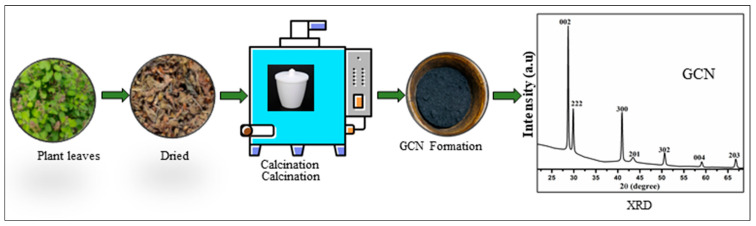
Overview of plant-based GCN nanoparticles formation.

**Table 1 pharmaceuticals-18-00820-t001:** Advantages and Drawbacks of Physical and Chemical Methods.

Methods	Advantages	Disadvantages	Refs.
Physical Methods			
Ball milling	Low cost	Agglomeration of nanoparticles with low production	[68,69,70,71,72]
Evaporation	Large production of nanoparticles	Costly; Time-consuming; high operating temperature	[70]
Arc discharge	Found high purity of nanoparticles; large production	Large size distribution of nanoparticles	[70,71,72,73]
Laser ablation	Found small size of nanoparticles; narrow size distribution	Very expensive, large amount of energy used; Less production of nanoparticles	[70]
Spray pyrolysis	Found high purity of nanoparticles	Require very high operating temperature	[71,72]
**Chemical methods**			
Sol-gel process	High-purity nanoparticles synthesized	Limited industrial applications; costly precursors; time-consuming process; difficult to handle	[77]
Chemical vapor deposition	Single deposition step is required for nanoparticles synthesis	High process cost; difficult to handle; Low yield	[78]
Reverse micelle	Simple process and easy to handle	Nanoparticle’s production is less with large size; toxic and hazardous chemicals are used as a stabilizing agent	[79]

**Table 2 pharmaceuticals-18-00820-t002:** Summarized the UV-Vis absorption reported work on Ag-NPs, Au-NPs, and ZnO-NPs from different parts of the plant for biomedical applications.

Plant Species	Absorption (nm)	Refs.
Ag Nanoparticles		
*Euphorbia serpens* Kunth	420	[2]
*Areca catechu* nut	415	[21]
*Carica papaya*	419	[92]
*Ferula asafoetida*	410–420	[101]
*Ipomoea carnea* Jacq.	390–410	[108]
Au Nanoparticles		
*Crataegus oxyacantha*	520–530	[1]
*Moringa oleifera*	535	[6]
Flaxseed	540	[7]
*Curcuma pseudomontana*	542	[9]
*Spondias dulcis*	536	[49]
ZnO Nanoparticles		
*Phlomis* Leaf	360	[29]
*Cymbopogon citratus*	370	[54]
*Mentha piperata*	385	[56]
*Rivina humilis*	370	[131]

**Table 3 pharmaceuticals-18-00820-t003:** Summarized the reported work on Ag-NPs from different parts of the plant for biomedical applications.

Plant Species	Part of Plant	Size/Shape	Application	Ref.
*Euphorbia serpens*	Leaves	~30–80 nm (Spherical)	Antimicrobial and Antibiofilm Activities	[2]
*Lawsonia inermis*	Leaves	~40 nm (Spherical)	Antimicrobial Activity	[3]
*Eucalyptus globulus* and *Salvia officinalis*	Leaves	~17.5 ± 5.89 nm and 34.3 ± 7.76 nm (Spherical)	Antioxidant and Antimicrobial Activities	[4]
*Aerva lanata*	Flower	~7 ± 3 nm (Spherical)	Antibacterial and Antioxidant Activities	[17]
*Areca catechu* nut	Fruit	~15–20 nm (Regular spherical)	Antioxidant and Antibacterial Activities	[21]
*Abelmoschus esculentus*	Flower	~5.52 to 31.96 nm (Spherical)	Antimicrobial Activity	[27]
*Acacia cyanophylla*	Leaves, Flowers and stems	~88.11 nm (Spherical)	Antibacterial Activity	[28]
*Allium cepa* L.	Bulb	~19.47 ± 1.12 nm (spherical)	Antioxidant, Antipathogenic, Anticholinesterase	[30]
*Eupatorium adenophorum*	Leaves	~117.75 nm (Spherical)	Antioxidant and Antibacterial Activities	[33]
*Withania coagulans*	seeds	~25 nm (spherical)	Antioxidant Activity	[35]
*Camellia sinensis*	Leaves	~8–26 nm (Spherical)	Antibacterial Activity	[37]
*Ocimum basilicum* L.	Leaves and stem	~35 nm (Oval)	Anticancer Activity	[38]
*Moringa oleifera*	Leaves	~25.235 ± 0.694 nm (spherical)	Antibacterial Activity	[40]
*Annona muricata*	Leaves	~19.63 ± 3.7 nm and 16.56 ± 4.1 nm	Antitumor Activity	[43]
Banana	Pulp	~42.97 nm (Spherical)	Antibacterial and Bioelectricity generation Activities	[55]
*Cupressus macrocarpa*	Leaves	~13.5–25.8 nm (Spherical)	Antibacterial Activity	[71]
*Aloe vera*	Leaves	~30–80 nm (spherical)	Antibacterial Activity	[83]
*Origanum majorana* L.	Leaves	~72.01 nm (Spherical)	Antioxidant Activity	[84]
*Carica papaya*	Leaves	10–25 nm (Round)	Anticancer Activity	[92]
*Cassia tora*	Seed	~50–60 nm (spherical)	Antibacterial Activity	[95]
*Eugenia roxburghii*	Leaves	~19–39 nm (Spherical)	Antimicrobial Activity	[96]
*Euphorbia hirta*	Leaves	~90–120 nm (spherical)	Antimicrobial Activity	[99]
Flaxseed	Seed	~46.98 ± 12.45 nm (Spherical)	Antibacterial and Antioxidant Activity	[100]
*Ferula asafoetida*	Leaves	~10 ± 2.77 nm (Circular)	Antibacterial and Cytotoxicity Activities	[101]
*Ocimum tenuiflorum*	Leaves	~10–65 nm (Round)	Helicoverpa Armigera Activity	[102]
Green Tea	…….	~50 nm (spherical)	Antimicrobial Activity	[103]
Grape seed	Seed	~10 to 30 nm (spherical)	Antimicrobial Activity	[105]
*Heteropyxis natalensis*	Leaves	~5–60 nm (spherical)	Antibacterial Activity	[106]
*Hibiscus rosasinensis*	Leaves, Flower, and Bark	~200 nm to 1 μm (Spherical)	Antimicrobial Activity	[107]
*Ipomoea carnea* Jacq.	Leaves	~11.21–46.90 nm (Spherical)	Antimicrobial Activity	[108]
*Jasminum nudiflorum*	Flower	~13 nm (Spherical)	Antioxidant and Antifungal Activities	[109]
*Morinda lucida*	Leaves	~11 nm (spherical)	Antioxidant and Antimicrobial Activities	[110]
*Mussaenda frondosa*	Leaves	~30–60 nm (spherical)	Antioxidant Activity	[111]
*Mikania cordata*	Leaves	~46–50 nm (spherical)	Antioxidant, Antimicrobial, Cytotoxic Activities	[113]
*Curcuma longa*	Leaves	~5–25 nm (Spherical)	Antibacterial Activity	[114]
*Ocimum canum*	Leaves	~15.76 nm (Rod and Spherical)	Antibacterial Activity	[115]
*Ocimum americanum*	Leaves	~48.25 nm (spherical)	Antibacterial, Antioxidant, and Anticancer Activities	[116]
*Otostegia persica*	Leaves	~36.5 ± 2.0 nm (Spherical)	Antibacterial, Antifungal, and Anti-inflammatory Activities	[117]
*Pisonia alba* L.	Leaves	Not mentioned (Spherical)	Antioxidant Activity	[118]
*Retama monosperma*	Root	~9.87–21.16 nm (Spherical)	Antimicrobial Activity	[121]
*Rosa canina*	Fruit	~13–31 nm (Spherical)	Antioxidant, Antimicrobial, and DNA cleavage	[122]
*Spilanthes acmella*	Flower	~10–35 nm (Spherical and oval)	Antioxidant Activity	[123]
*Sesbania grandiflora*	Leaves	~10–25 nm (Granular-like)	Antibacterial and Antioxidant Activities	[124]
*Sambucus ebulus*	Leaves	~18.6 nm (Spherical)	Antioxidant and Antibacterial Activities	[125]
*Thuja orientalis*	Leaves	~85.77 nm (spherical)	Antimicrobial Activity	[126]
*Terminalia chebula*	Fruit	~10–30 nm (Spherical)	Antioxidant, Protein Leakage Analysis Antibacterial, Zebrafish Embryonic Toxicology	[127]
*Tropaeolum majus*	Leaves	~25 nm (Crystalline)	Antimicrobial Activity	[128]
*Aloe vera*	Stems	Cubical, Spherical, and Triangles	Antibacterial Activity	[137]

**Table 4 pharmaceuticals-18-00820-t004:** Summarized the reported work on Au-NPs from different parts of the plant for biomedical applications.

Plant Species	Part of Plant	Size/Shape	Applications	Ref.
*Crataegus oxyacantha*	Twig	~85 nm (Spherical)	Urease inhibitory Activities	[1]
*Moringa oleifera*	Leaves	~14–30 nm	Antibacterial, Antioxidant, and Cytotoxicity Activities	[6]
Flaxseed	Seed	~3.4–6.9 nm (Spherical and Triangular) 10–30 Spherical	Anticancer Activity	[7]
Oak gum	Fruit	~10–15 nm (Spherical)	Antioxidant and Anti-colon cancer Activities	[8]
*Curcuma pseudomontana*	Flower	~20 nm (Spherical)	Antimicrobial, antioxidant, anti-inflammatory activities	[9]
*Glaucium flavum*	Leaves	~32 nm (Hexagonal, Triangular, and Spherical)	Anticancer Activity	[15]
*Jatropha integerrima* Jacq.	Flower	~38.8 nm (Spherical)	Antibacterial activity	[20]
*Licorice*	Root	~ 2.647 nm to 16.25 nm (Circular)	Antimicrobial and Anticancer Activities	[25]
*Kaempferia parviflo*	Rhizomes	~ 44 ± 3 nm (Spherical)	Antimicrobial and Antioxidant Activities	[26]
*Citrus* peel	Fruit	~13.65–16.80 nm (Spherical)	Anti-inflammatory activity	[32]
*Cynodondactylon*	Grass	~21.33 nm (Spherical and Irregular)	Cytotoxicity and Antibacterial activities	[39]
*Spondias dulcis*	Fruit	~36.75 ± 11.36 nm (Spherical)	Cytotoxic activity in human breast cancer cells	[49]
*Saffron*	Flower	~25 nm (Spherical and Oval) 15–50 Spherical	Antioxidant and Cytotoxicity Activities	[51]
*Delphinium chitralense*	Tuber	~100–300 nm (Cubic)	Enzyme inhibitory Activity	[82]
*Combretum erythrophyllum*	Leaves	~13.20 nm (Spherical)	Antibacterial, Cell viability Activities	[85]

**Table 5 pharmaceuticals-18-00820-t005:** Summarized the reported work on ZnO-NPs from different parts of the plant for biomedical applications.

Plant Species	Part of Plant	Size/Shape	Applications	Ref.
*Borreria hispida*	Leaves	~21.87 nm (Hexagonal)	Antioxidant Activity	[10]
*Ailanthus altissima*	Leaves	~13.27 nm (Spherical)	Antibacterial and Antioxidant Activities	[11]
*Passiflora subpeltata*	Leaves	~40–50 nm (Irregular)	Antibacterial Activity	[12]
*Andrographis alata*	Whole plant	~45 ± 4.23 nm (Spherical, Oval, and Hexagonal)	Antibacterial, Antioxidant, Antidiabetic, and Anti-Alzheimer Activities	[13]
*Mucuna pruriens*	Peel	~21.60 to 47.16 nm (Spherical)	Anticancer, antioxidant activity	[14]
*Bergenia ciliata*	Rhizome	(Flower-like bundles)	Antibacterial, anticancer potential	[16]
*Berberis aristata*	Leaves	~5–25 nm (Needle)	Antibacterial activities and Antioxidant	[23]
*Lantana camara*	Flower	~25 nm (Spherical)	anti-inflammatory	[24]
*Phlomis* leaf	Leaves	~79 nm (Hexagonal)	Cytotoxicity and Antibacterial Activities	[29]
*Lepidium sativum*	Seeds	~36.96–44.50 (Spherical)	Anticancer activity	[31]
Plantain peel	Peel	~20 nm (Spherical)	Antibacterial Activity	[34]
*Camellia sinensis*	Leaves	~6 to 112 nm (Spherical)	Antioxidant, Antibacterial, and Anticancer Activity	[42]
*Caesalpinia crista*	Seed	~34.67 nm (Irregular)	Antibacterial, Antioxidant, and Anticancer Activities	[44]
*Aquilegia pubiflora*	Leaves	~34.23 nm (Spherical or Elliptical)	Antiproliferative Activity	[45]
*Amygdalus scoparia*	Bark	~15–40 nm (Spherical)	Antimicrobial, Anticancer, and Antidiabetic	[46]
*Sauropus androgynus*	Leaves	~12 to 23 nm (Spherical)	Antineoplastic Activity	[50]
*Capparis spinosa*	Fruit	~37.49 nm (Spherical)	Antioxidant Activity	[52]
*Ipomoea Sagittifolia* Burm.f	Leaves	~51.2 ± 8.5 nm (Hexagonal)	Antibacterial, Antioxidant, and Anticancer Activities	[53]
*Cymbopogon citratus*	Aerial Part	~21 nm (Spherical)	Antimicrobial and Anticancer Activity	[54]
*Mentha piperata*	Leaves	15 to 27 nm (Globular and Oblong)	UTI-resistant pathogens	[56]
*Mangifera indica*	Seed	~40–70 nm (Cylindrical)	Antibacterial and Antioxidant Activities	[65]
Olive fruit	Fruit	~56.8 ± 0.6 (Spheroidal)	Antioxidant Activity	[86]
*Punica granatum*	Peel	~20–40 nm (Spherical and Hexagonal)	Antimicrobial Activity	[90]
*Myrica esculenta*	Fruit	~115 ± 3.21 nm (Pellets-like)	Antioxidant, antimicrobial, photocatalytic	[93]
*Capsicum chinense*	Fruit	~12.7 nm (Spherical)	Cytotoxicity and Antioxidant Activities	[94]
*Myristica fragrans*	Fruit	~43.3 to 83.1 nm (Semi spherical)	Antioxidant, antimicrobial, photocatalytic	[97]
Boerhaavia erecta	Leaves	~20.55 nm (Spherical)	Antioxidant and Antiglycation Activities	[104]
*Brassica oleracea*	Leaves	~52 nm (Flower-like)	Antimicrobial and Larvicidal Activities	[112]
*Punica granatum*	Peel	~10.45 nm (Spherical)	Antimicrobial Activity	[130]
*Rivina humilis*	Leaves	~14.4 nm (Irregular and Circular)	Anticancer, Cytotoxicity, and Antiproliferative Activities	[131]
*Sphagneticola trilobata*	Leaves	~29.83 (Spherical)	Colon cancer and Antioxidant Activities	[132]
*Thryallis glauca*	Leaves	~50 nm (Hexagonal wurtzite)	Antioxidant and Antibacterial	[133]
*Thuja officinalis*	Leaves	4–5 nm	Antimicrobial	[134]
*Ocimum tenuiflorum*	Leaves	~30 to 40 nm Flakes	Photocatalytic-Antimicrobial Activity	[136]
*Ricinus Communis* L.	Fruit	Hexagonal irregular	Antibacterial Activity	[138]

**Table 6 pharmaceuticals-18-00820-t006:** Comparison: Traditional Methods between Nanoparticle-Based Methods in Biological Applications.

Application	Traditional Methods	Nanoparticle-Based Method	Advantages of Nanoparticles	Refs.
Cancer Treatment	Chemotherapy; radiotherapy	Targeted drug delivery; photothermal therapy	Minimize damage to healthy cells	[16,49,92,132]
Antibacterial	Broad-spectrum antibiotics	Ag-, ZnO-, Cu-NPs etc.	Effective against resistant strains	[2,27,138,154]
Biosensors	Enzyme and antibody-based detection	Nanoparticles-based sensors	Faster detection	[5,13,55,60]
Wound healing	Traditional dressing; antiseptic solution	Nanoparticles-embedded dressing such as Ag-, ZnO-NPs	Antibacterial, faster healing, reduced infection rate	[24,53,138]
Drug delivery	Oral intravenous drug	Targeted delivery	Site specific action, reduce side effects	[57,58]

## Data Availability

The data presented in this study are included within the article.

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
