# Peer review of "A Review on Biomedical Applications of Plant Extract-Mediated Metallic Ag, Au, and ZnO Nanoparticles and Future Prospects for Their Combination with Graphitic Carbon Nitride"

_pharmaceuticals, 2025, doi:10.3390/ph18060820_

Round 1

Reviewer 1 Report

Comments and Suggestions for Authors

The article outlines methods for metal nanoparticle synthesis using plant extracts as an alternative to traditional chemical compounds in the synthesis process. Particularly, the plant extracts are used to reduce metal ions to their atomic form, creating nanoparticles. The review covers a lot of material and it is clear that the authors have put a lot of work into its preparation. Green synthesis of nanoparticles is a topic which deserves much attention given modern shift toward environmentally friendly technologies. I believe that the review would be of interest to researchers who are looking for a concentrated summary of previously published results in this field. However, there are some aspects of the article which can be improved, both in terms of its content (what themes and questions are covered in the review) and in terms of its presentation (how the text is structured). Below I list my thought on how the article can be improved.

Content improvements:

1.1 I believe it would be better if the Highlights section stated some key takeaways or conclusions that could be interesting to the reader, rather than just an outline of the article

1.2 DLS is not mentioned in the article, even though it is a widely used nanoparticle size characterization technique. Is not used for particles synthesized with methods described in this review? Are there some limitations on what methods can be used for this kind of materials?

1.3 The review provides many lab-scale examples of green synthesis of nanoparticles, however scaling of that process is not discussed. It would be interesting to understand if the described techniques could be used on the industrial scale. I believe readers would benefit if the authors discussed this issue or at least referred the reader to relevant literature (for example: 10.1007/s10311-012-0395-x, 10.1016/j.sajce.2021.06.008, 10.1016/j.cep.2021.108439).

1.4 I believe that the drawbacks of the green-synthesis approach compared to traditional methods could be better highlighted in this review to make it more objective and allow the reader to see the current challenges that this approach is experiencing (for example, 10.1016/j.eti.2022.102336).

1.5 The review showcases many examples of different plants being used for particle synthesis. However, it never tries to compare how the choice of plant affects the results of the synthesis. Is it really important which plant is chosen? Have any comparative studies been conducted between different plants? Has the variability of the results when using the same plant but harvested at a different time or location been evaluated?

1.6 Would it be beneficial if the researchers agreed on using the same plant or a few model plants with well-defined characteristics for the synthesis? For example, in the area of toxicity testing a few well characterized model organisms exist and usually researchers use those organisms in their experiments. Do such model plants exist when it comes to green nanoparticle synthesis? If so, the review could list them and justify why exactly those plants are used most often.

1.7 When talking about biomedical applications of nanoparticles, the authors could mention if any nanoparticles prepared using green synthesis have undergone clinical trials in the past or if any promising drugs in the preclinical stage exist. If no such advancements exist, it would be interesting if the authors attempted to identify why that is the case.

1.8 Additionally, in the context of biomedical application of nanoparticles, it is great that some potential uses have been demonstrated, however it is far more important to determine whether those uses have any advantages compared to currently employed techniques. I believe the review would be much more comprehensive if that aspect was considered.

Presentation improvements:

2.1 In many parts of the text where some process or classification is described (e.g. lines 140-160, 439 and others), it might be more accessible for the readers if the material is presented in a structured manner (for example, as a multi-level list) rather than plain text.

2.2 In some cases when results of other studies are cited (such as particle sizes on lines 254-266) it would be easier to understand the presented data if it was formatted as a table.

2.3 Statement at lines 172 - 173 requires a source and could benefit from an example.

2.4 It is not very clear what the wavelengths specified on lines 192-194 mean. I think additional explanation would be appropriate here. Also, I would suggest moving this information into a table and then just referring to the table in the text rather than listing all the values.

2.5 Text formatting can be improved. For example some words are printed in bold with no clear meaning (e.g. lines 184, 204, etc).

2.6 Zeta potential of nanoparticles depends strongly on the pH of the medium and presence of ions in it. Hence when citing zeta potential values of nanoparticles (e.g. lines 276-278) it would be helpful to specify under what conditions it was measured.

2.7 On lines 380-381 when IC50 value for the DPPH assay is presented only the particle concentration is given, however the IC50 value would also depend on the DPPH concentration used in the assay.

2.8 When particle size is provided (for example, in table 3) it could be helpful to specify which method was used for determining the size. First of all it would allow the readers to see if the data comes from a single or from multiple different methods and also evaluate whether results obtained form different methods are in agreement.

2.9 Quality of figures can be improved as some are presented in low resolution

Comments on the Quality of English Language

The language is generally clear, however some sentences are overloaded when a large number of items are listed in a single sentence. I believe in such cases it would be better to present listings as tables and refer to them in the text instead.

Author Response

Reviewer #1

The article outlines methods for metal nanoparticle synthesis using plant extracts as an alternative to traditional chemical compounds in the synthesis process. Particularly, the plant extracts are used to reduce metal ions to their atomic form, creating nanoparticles. The review covers a lot of material and it is clear that the authors have put a lot of work into its preparation. Green synthesis of nanoparticles is a topic which deserves much attention given modern shift toward environmentally friendly technologies. I believe that the review would be of interest to researchers who are looking for a concentrated summary of previously published results in this field. However, there are some aspects of the article which can be improved, both in terms of its content (what themes and questions are covered in the review) and in terms of its presentation (how the text is structured). Below I list my thought on how the article can be improved.

Content improvements:

Comment #1.1:I believe it would be better if the Highlights section stated some key takeaways  or conclusions that could be interesting to the reader, rather than just an outline of the article.

 Reply: Thank you so much for your suggestion. As per the direction we have modified the outline in the first page.

Comment#2.1: DLS is not mentioned in the article, even though it is a widely used nanoparticle size characterization technique. Is not used for particles synthesized with methods described in this review? Are there some limitations on what methods can be used for this kind of materials?

Reply: We thank reviewer for bringing this requirement to our attention. Now we added DLS (line 335-342), please see figure VI(II) (g, h, i). Additionally, we mentioned the limitation at the end of the morphology and size distribution studies.

Comment #3.1: The review provides many lab-scale examples of green synthesis of nanoparticles, however scaling of that process is not discussed. It would be interesting to understand if the described techniques could be used on the industrial scale. I believe readers would benefit if the authors discussed this issue or at least referred the reader to relevant literature (for example: 10.1007/s10311-012-0395-x, 10.1016/j.sajce.2021.06.008, 10.1016/j.cep.2021.108439).

 Reply: Thank you for the valuable comment. We have added a brief discussion on the  challenges and prospects of scaling green synthesis methods for industrial applications, along with references (lines 609-610 and 628-633).

Comment #4.1: I believe that the drawbacks of the green-synthesis approach compared to traditional methods could be better highlighted in this review to make it more objective and allow the reader to see the current challenges that this approach is experiencing (for example, 10.1016/j.eti.2022.102336).

Reply: Thank you so much for your suggestion. As per the direction, we have modified the manuscript and added a part that highlights the drawbacks (6. Limitations of Green Synthesized nanoparticles) (Lines 604-633).

Comment #5.1: The review showcases many examples of different plants being used for particle synthesis. However, it never tries to compare how the choice of plant affects the results of the synthesis. Is it really important which plant is chosen? Have any comparative studies been conducted between different plants? Has the variability of the results when using the same plant but harvested at a different time or location been evaluated?

 Reply: Thank you for your comment. We have added a brief discussion mentioning that  plant choice, harvest time, and location can affect nanoparticle synthesis (lines  603-608.

Comment#6.1:Would it be beneficial if the researchers agreed on using the same plant or a few model plants with well-defined characteristics for the synthesis? For example, in the area of toxicity testing a few well characterized model organisms exist and usually researchers use those organisms in their experiments. Do such model plants exist when it comes to green nanoparticle synthesis? If so, the review could list them and justify why exactly those plants are used most often.

 Reply: We thank the reviewer for the important comment. It is true that the literature is very  rich, but to the best of our knowledge, no pattern to identify the best plant-extract to  be use for specific synthesis has not been yet identified. Maybe the use of tool like machine learning process could be useful to identify patterns in data or make  predictions for specific plants with well-defined characteristics for targeted synthesis. We pointed out this fact in the abstract and in the conclusion (line 640-645 and 657-660) to open the discussion and highlight that such future investigations are of interest in the field.

Comment #7.1: When talking about biomedical applications of nanoparticles, the authors could mention if any nanoparticles prepared using green synthesis have undergone clinical trials in the past or if any promising drugs in the preclinical stage exist. If no such advancements exist, it would be interesting if the authors attempted to identify why that is the case.

Reply: Thank you so much for your suggestion. We have included a brief mention on the current status of green-synthesized nanoparticles in preclinical and clinical studies at the end of the introduction with two recent references (lines 91-93).

Comment #8.1:Additionally, in the context of biomedical application of nanoparticles, it is great that some potential uses have been demonstrated, however it is far more important to determine whether those uses have any advantages compared to currently employed techniques. I believe the review would be much more comprehensive if that aspect was considered.

 Reply: Thank you so much for your suggestion. Now we added a comparison, please see table 6.

Presentation improvements

Comment #2.1: In many parts of the text where some process or classification is described (e.g., lines 140-160, 439 and others), it might be more accessible for the readers if the material is presented in a structured manner (for example, as a multi-level list) rather than plain text.

 Reply: Thank you so much for your suggestion. As per the direction we revised the manuscript.

Comment #2.2:In some cases when results of other studies are cited (such as particle sizes on lines 254-266) it would be easier to understand the presented data if it was formatted as a table.

 Reply: Thank you so much for your feedback. We have removed those lines, as the  information is already presented in 

               Tables  3,4,5.

Comment #2.3: Statement at lines 172-173 requires a source and could benefit from an example.

  Reply: Thank you so much for your suggestion. We added example and source.

Comment #2.4: It is not very clear what the wavelengths specified on lines 192-194 mean. I think additional explanation would be appropriate here. Also, I would suggest moving this information into a table and then just referring to the table in the text rather than listing all the values.

 Reply: Thank you so much for your suggestion. As per the direction, we have added a table (See Table-1).

Comment #2.5:Text formatting can be improved. For example, some words are printed in bold with no clear meaning (e.g., lines 184, 204, etc).

  Reply: We apologize for the same and thank you for bringing this requirement to our attention. Now we corrected all errors.

Comment #2.6: Zeta potential of nanoparticles depends strongly on the pH of the medium and presence of ions in it.  Hence when citing zeta potential values of nanoparticles (e.g., lines 276-278) it would be helpful to specify under what conditions it was measured.

 Reply: Thank you so much for your suggestion. As per the direction we have added pH  values (line 333).

Comment #2.7: On lines 380-381 when IC50 value for the DPPH assay is presented only the particle concentration is given, however the IC50 value would also depend on the DPPH concentration used in the assay.

Reply: Thank you for the comment. We have now corrected (line 452).

Comment #2.8: When particle size is provided (for example, in table 3) it could be helpful to specify which method was used for determining the size. First of all, it would allow the readers to see if the data comes from a single or from multiple different methods and also evaluate whether results obtained from different methods are in agreement.

Reply: Thank you for your insightful comment. We would like to clarify that the different  techniques used to study each type of nanoparticle have already been included in the tables  3,4,5. To address this, we have added a brief note in the morphology and size distribution  section

Comment #2.9:Quality of figures can be improved as some are presented in low resolution.

Reply: We thank reviewer for bringing this requirement to our attention, the manuscript is now revised.

Comments on the Quality of English Language

Comment #2.9: The language is generally clear; however, some sentences are overloaded when a large number of items are listed in a single sentence. I believe in such cases it would be better to present listings as tables and refer them in the text instead.

Reply: We thank reviewer for bringing this requirement to our attention, it is now corrected.

Reviewer 2 Report

Comments and Suggestions for Authors

Dear Authors,

Congratulations on the clear and well-written manuscript. These are some points for your consideration and revision. 

Ensure that the manuscript's title and abstract indicate that it is a review article.

In the highlights, specify that this manuscript is a review article. Additionally, summarise the main topics and findings reviewed in the manuscript.

For your paragraph from line 43 to 87, it is too long. Break this paragraph up for an easier read.

In section 2, break the top-down approach and bottom-up approach into two paragraphs.

In section 3.1, maybe you would like to consider putting the UV-VIS absorption spectra into a table form for a summary.

Check all the buildings to make sure that it is consistent.

Is section 4 a repeat of what was mentioned earlier? Maybe the title can be changed?

In section 4.3.2, break up the long paragraph into different paragraphs.

Ensure that all figures are in the right ratio.

Please check that all the images in Figure 8 are original.

Author Response

                                                                   Reviewer Two                               

Congratulations on the clear and well-written manuscript. These are some points for your consideration and revision.

Comment #1:Ensure that the manuscript's title and abstract indicate that it is a review article.

Reply: We thank reviewer for bringing this requirement to our attention, now, it is corrected.

Comment #2: In the highlights, specify that this manuscript is a review article. Additionally, summaries the main topics and findings reviewed in the manuscript.

Reply: Thank you for this useful suggestion. We have improved the highlights to enhance their clarity about the manuscript content.

Comment #3: For your paragraph from line 43 to 87, it is too long. Break this paragraph up for an easier read.

Reply: Thank you for this useful suggestion. It is now corrected.

Comment #4: In section 2, break the top-down approach and bottom-up approach into two paragraphs.

Reply: As per the suggestion, now we are corrected.

Comment#5: In section 3.1, maybe you would like to consider putting the UV-VIS absorption spectra into a table form for a summary.

Reply: Thank you for this useful suggestion now revised and highlighted. Please see Table-1.

Comment #6: Check all the buildings to make sure that it is consistent.

        Reply:  We have carefully reviewed all references to the buildings throughout the   manuscript to ensure consistency. Necessary corrections have been made to  maintain clarity and uniformity.

Comment #7: Is section 4 a repeat of what was mentioned earlier? Maybe the title can be changed?

Reply: We thank reviewer for bringing this requirement to our attention, it is now corrected and the section 4 is entitled “Literature study”.

Comment #8: In section 4.3.2, break up the long paragraph into different paragraphs.

         Reply: Thank you for this useful suggestion now revised and highlighted

Comment #9: Ensure that all figures are in the right ratio.

  Reply: Thank you very much for your valuable suggestions. We have checked the figure and pay attention that they exhibit the right ratio.

Comment #10: Please check that all the images in Figure 8 are original.

              Reply: Yes, in figure 8 all figures are original.

Reviewer 3 Report

Comments and Suggestions for Authors

The aim of the “Biomedical Applications of Plant Extract-Mediated Metallic Ag, Au, and ZnO Nanoparticles and Future Prospects for their combination with Graphitic Carbon Nitride” paper is very well explained and comprehensive, showing the great work that the authors put into it.

Also, your paper is well-organised and interesting, presenting plenty of data.

Please, check below some of my suggestions/comments:

-Please, insert a short phrase at the end of the introduction section to increase the scope of the research.

-In Section 2.2, make a short table highlighting the most relevant advantages and drawbacks of biological and chemical methods.

-Please, move some of the paragraphs after each table or figure, so the section won’t end with just a figure/ table (figure 1, 2, 5, 6, 8 & table 1, 2, 3).

-Figure 7 is very hard to read. Could the authors increase its resolution?

Author Response

                                                                   Reviewer 3

The aim of the “Biomedical Applications of Plant Extract-Mediated Metallic Ag, Au, and ZnO Nanoparticles and Future Prospects for their combination with Graphitic Carbon Nitride” paper is very well explained and comprehensive, showing the great work that the authors put into it.

Also, your paper is well-organized and interesting, presenting plenty of data. Please, check below some of my suggestions/comments:

Comment #1: Please, insert a short phrase at the end of the introduction section to increase the scope of the research.

 Reply: Thank you for this useful suggestion now we added some lines at the end of the introduction section. We have increased the scope of the research and extended it to highlight the need for future investigation using machine learning processes.

Comment #2: In Section 2.2, make a short table highlighting the most relevant advantages and drawbacks of biological and chemical methods.

 Reply: Thank you very much for your valuable suggestions, a table is now added (table 1)

Comment #3: Please, move some of the paragraphs after each table or figure, so the section won’t end with just a figure/ table (figure 1, 2, 5, 6, 8 & table 1, 2, 3).

Reply: We thank reviewer for bringing this requirement to our attention, it is now corrected.

Comment #4: Figure 7 is very hard to read. Could the authors increase its resolution?

 Reply: Thank you for bringing this to our attention. The resolution and size of the figure 7 was increased.